# Core Temperature during Cold-Water Triathlon Swimming

**DOI:** 10.3390/sports9060087

**Published:** 2021-06-20

**Authors:** Lars Øivind Høiseth, Jørgen Melau, Martin Bonnevie-Svendsen, Christoffer Nyborg, Thijs M. H. Eijsvogels, Jonny Hisdal

**Affiliations:** 1Department of Anesthesiology, Division of Emergencies and Critical Care, Oslo University Hospital, 0450 Oslo, Norway; 2Institute of Clinical Medicine, University of Oslo, 1171 Oslo, Norway; jorgen@melau.no (J.M.); nyborgchristoffer@gmail.com (C.N.); jonny.hisdal@medisin.uio.no (J.H.); 3Department of Vascular Surgery, Oslo University Hospital, 0586 Oslo, Norway; martin.bonnevie@gmail.com; 4Prehospital Division, Vestfold Hospital Trust, 3116 Tønsberg, Norway; 5Radboud Institute for Health Sciences, Department of Physiology, Radboud University Medical Center, 6525 Nijmegen, The Netherlands; thijs.eijsvogels@radboudumc.nl

**Keywords:** core temperature, swimming, triathlon, wetsuit

## Abstract

Triathlon and other endurance races have grown in popularity. Although participants are generally fit and presumably healthy, there is measurable morbidity and mortality associated with participation. In triathlon, most deaths occur during the swim leg, and more insight into risk factors, such as hypothermia, is warranted. In this study, we measured the core temperature of 51 participants who ingested temperature sensor capsules before the swim leg of a full-distance triathlon. The water temperature was 14.4–16.4 °C, and the subjects wore wetsuits. One subject with a low body mass index and a long swim time experienced hypothermia (<35 °C). Among the remaining subjects, we found no association between core temperature and swim time, body mass index, or sex. To conclude, the present study indicates that during the swim leg of a full-distance triathlon in water temperatures ≈ 15–16 °C, subjects with a low body mass index and long swim times may be at risk of hypothermia even when wearing wetsuits.

## 1. Introduction

Endurance races, such as marathons and triathlons, emerged during the 1980s and have since become more popular. Although participants are generally fit and healthy, previous studies reported sudden death occurring in 0.5 to 1.7 athletes per 100,000 participants [1,2]. Most deaths in triathlon occur during the swim leg and appear to be related to cardiovascular disease [3]. The mortality rates reported above underscore the need to take safety precautions and to reduce the chance of risk factors such as hypothermia occurring during the swim leg. Knowledge about such risks is also relevant to other events involving open-water swimming.

While at rest, the thermoneutral water temperature is approximately 35.0–35.5 °C [4]. Immersion in cold water may lead to hypothermia, which is generally defined as a body core temperature < 35 °C. Swimming induces a complex interplay between metabolic heat production, heat loss to the water, and heat transfer through conduction and convection within the body, which affects the core temperature. It is generally believed that subcutaneous fat insulates from heat loss [5,6]. Increased muscle activity may raise the core temperature through metabolic heat production but may also increase heat loss to the water, as there is a more uniform temperature distribution with increased muscle blood flow, reducing the insulating effect of otherwise hypoperfused tissue [4]. During swimming in cold water, the ability to swim may deteriorate, even in the absence of central hypothermia [7].

When the Norseman Xtreme Triathlon was first organized in 2003, it spawned a new triathlon trend. The event starts at sea level, with a 3800 m swim in the Hardanger Fjord, continues with a 180 km cycling leg peaking at 1200 m above sea level, and finishes with a 42.2 km run leg, starting at 190 m above sea level and finishing at 1880 m above sea level, at the Gaustatoppen mountain peak. Over the years, similar competitions have popped up around the world, becoming increasingly popular. Since 2012, an XTRI-World tour has been organized, comprising 17 triathlons around the world. Several of these, such as the Swedman and Celtman events, take place in cold locations, with correspondingly cold water during the swim leg.

Although Norseman is a summer event, water and air temperatures are often low. Participants enter the water at approx. 04:45 a.m. to position for the 05:00 a.m. start of a swim leg with a cut-off time set to 135 min, with the fastest swimmers finishing in approximately 50 min. In 2015, with the water temperature at approximately 10 °C, the Norseman organizers decided to shorten the swim for safety reasons due to the cold water. This situation led to the acknowledgment of a lack of data on which to make qualified decisions regarding the risk of hypothermia during such races.

The aim of the present study is to estimate the incidence of hypothermia during the swim leg of a cold-water full-distance triathlon by measuring core temperature during the 2017, 2018, and 2019 competitions. As a secondary purpose, we also studied the first hour of cycling following the swim leg to evaluate a possible afterdrop.

## 2. Materials and Methods

The study was evaluated by the Regional Ethical Committee (helseforskning.etikkom.no; REK references 2015/1533 and 2017/1138) and the Data Protection Officer at Oslo University Hospital (2017/8299). The study was conducted on participants in the Norseman Xtreme Triathlon in 2017 (n = 16), 2018 (n = 22), and 2019 (n = 13). Subjects were recruited via social media and included after written informed consent. Exclusion criteria were a history of gastrointestinal disease or surgery, implanted medical devices, a scheduled MRI scan within seven days after the race, pregnancy, or weight below 36.5 kg, corresponding to the list of contraindications listed by the manufacturer of the ingestible temperature sensors used in the study.

Core temperature was recorded with ingestible temperature sensors (e-Celsius; BodyCAP, Hérouville Saint-Clair, France). Sensors were pre-programmed to a sampling frequency at 2 min intervals and ingested the night before the race. No specific instructions were given regarding pace, hydration, or fuel ingestion. After crossing the finish line, data from the temperature sensors were downloaded (e-Viewer Performance monitor; BodyCAP, Hérouville Saint-Clair, France). Water temperature was determined as the average of four readings during the swim (Fluke 51; Fluke Corporation, Everett, WA, USA). Air temperatures for 2017 and 2018 were provided by the Norwegian Meteorological Institute (Norwegian Meteorological Institute, 2019). Air temperature for 2019 was determined with a calibrated weather meter (Kestrel 5400 Heat Stress Tracker, Kestrel instruments, Boothwyn, PA, USA).

Temperature data were downloaded as csv files to Microsoft Excel [8] and handled in R 4.0.4 [9]/RStudio 1.4.1106 [10], using the Tidyverse packages [11]. Obviously, erroneous data from the swim (e.g., from ingestion of water) were removed manually. Multivariable linear regressions were performed, with the core temperature at the end of swim as the outcome variable, and swim time, body mass index (BMI), and sex as explanatory variables due to their presumed biological importance [1]. First, a full model with interaction effects was calculated, after which interaction effects and main effects with the highest *p*-values were removed. We then entered all observations during the swim in a linear mixed regression model with the subject as random effect and an AR(1) covariance structure due to the repeated measurements within subjects. Data were mean (SD) or median (25th, 75th percentiles), unless otherwise stated. *p*-values < 0.05 were considered statistically significant. Normality assumptions were checked using histograms, Q–Q plots, and by plotting fitted values vs. residuals.

## 3. Results

### 3.1. Participants

Seventy participants were recruited to participate in the study. However, data from 19 participants could not be downloaded after the race, leaving 51 participants (9 women and 42 men) for analysis. The participants’ demographics are presented in Table 1. Air temperature in the transition zone after the swim leg was 9.9 °C in 2017, 8.6 °C in 2018, and 15.6 °C in 2019.

### 3.2. Core Temperature during the Swim

Temperature measurements during the swim leg and the first hour after the swim leg are presented in Figure 1. Only one subject, in 2018, experienced hypothermia, defined as body temperature < 35 °C during the swim. This subject’s lowest temperature was 33.2 °C, which was measured after 128 min of swimming.

In the multivariable regression analysis, we found no statistically significant effect of BMI; 0.0096 °C/kg × m^−2^ (95% CI −0.15 to 0.16, *p* = 0.90), but significant main effects and interaction effects of swim time and sex (Table 2). These effects corresponded to an effect of time of −0.060 °C/min (95% CI −0.090 to −0.029, *p* < 0.001) for women and −0.012 °C/min (95% CI −0.037 to 0.014, *p* = 0.50) for men. The temperature at the end of the swim, swim time, sex, and BMI are presented in Figure 2.

One outlier had a long swim time (128 min) and a low BMI (18 kg/m^2^) (Figure 2). Except for this observation, none of the explanatory variables were statistically significant (Table 3). This was also found when entering all temperature observations during the swim with a first-order autoregressive (AR(1)) covariance structure due to repeated measures over time.

### 3.3. Core Temperature after the Swim

Temperature measurements after the swim leg were plotted, and a local polynomial regression-line (LOESS) smoothing line was added (Figure 3). As determined by visual inspection, there were no obvious signs of afterdrop at group-level, based on the LOESS-line.

## 4. Discussion

This study’s main finding was that only one subject experienced hypothermia during the cold-water swimming leg of a full-distance triathlon. After excluding this outlier, no significant association was found between swim time or BMI and body core temperature. Furthermore, the study did not detect any tendency for body temperature to change over time during the swim for the group as a whole.

The results of a 2018 study on core temperature measurements during a 20 min flume swim in different temperatures and wetsuit combinations prompted recommendations of minimum water temperatures during triathlons of 12 °C with wetsuits, and 16 °C without wetsuits [12]. Except for the outlier, our study does not clearly support the flume swim study’s finding that the leanest subjects were more susceptible to hypothermia. Another study from the same group regarding swimming for up to 2 h in 14–20 °C water temperatures without a wetsuit [13] led to making wetsuits mandatory in marathon swims at temperatures below 18 °C, and optional at temperatures of 18–20 °C. In this study also, the rate of change of body temperature was associated with skinfold thickness and heat production per body surface area. It is important to note that the swimmers had a poor perception of their own core temperature [13], possibly due to adaptation [14]. As in our study, sex did not seem to be an essential factor.

When considering the safety of a triathlon swim, associations and changes at the group level are less relevant, as care must be taken to curb the risk for single outliers. The perhaps most notable observation in the present study is related to the subject who experienced mild hypothermia during the swim, with the lowest measured temperature of 33.2 °C. This subject had the lowest BMI (18 kg/m^2^) as well as the longest swim time (134 min). The long swim time could have potentially contributed to hypothermia, both as a result of time in the water and of low metabolic activity due to slow swimming, compatible with a recommendation that open-water swimmers should not be thin and slow [4,15]. Combined with the insight of poor self-assessment of core temperature, this observation should be considered when organizing open-water swims.

A previous study of open-water swimming at approximately 10 °C, while wearing a wetsuit found highly individual core temperature responses [16]. As a result of one subject experiencing hypothermia, most subjects were limited to a 55 min swim. While no subjects experienced hypothermia during the first 30 min, extrapolation of the temperature trajectories indicated that a large proportion would be hypothermic after a 135 min swim. In the present study, we did not find an association between swim time and core temperature. When linearly extrapolating the temperature trajectory of the last 30 min of the swim a further 30 min, only two additional subjects were on a trajectory to temperatures below 35 °C. This was probably due to the relative warmth (14.4–16.4 °C) of the water during the three competitions and to wearing wetsuits. Unlike the previous study [16], the present study was performed in racing conditions, which may induce higher heat production due to increased intensity.

Afterdrop, i.e., a further reduction of core temperature after exiting the water, has been repeatedly reported [16,17] and is presumably caused by both conduction and convection [18]. We did not find clear evidence of a significant afterdrop on a group level, as judged by the LOESS-line. Interpretation of individual data was complicated by many apparently erroneously low values, presumably caused by ingestion of cold drinks after the end of the swim, and these data should be interpreted with caution. However, the absence of a clear afterdrop is consistent with the absence of a reduction in body temperature over time, again probably related to the relatively warm water. Unlike other long-distance triathlons, the Norseman Xtreme Triathlon permits its participants, i.e., the subjects of the present study, to wear neoprene socks and caps, which may contribute to reducing the participants’ core temperature drop. In Norseman, the temperature during the transition from swimming to cycling is often relatively cold, and afterdrop may therefore be of less relevance for the present study than for other studies.

In a large proportion (27%) of the subjects, temperature measurements could not be performed. The capsules containing temperature sensors were ingested the evening before the race, and in most of the unsuccessful cases, the capsule is likely to have passed through the gastrointestinal tract before the reading was performed. Ingesting the capsule immediately before the race could have reduced this problem, but we wanted the capsule to be passed through the ventricle into the intestines before the race. Several readings immediately after the swim showed abrupt drops in temperature, which were probably caused by drinking cold fluids. Some readings during the swim also showed apparently erroneous abrupt drops in temperature, possibly as a result either of ingesting seawater, or the capsule being close to the abdominal wall. The exclusion of these readings was based on a subjective judgment, which carries some uncertainty.

In the regression models, we started with a full model comprising the variables we considered biologically plausible (swim time, sex, and BMI), first excluding statistically non-significant interaction effects. In univariable analyses, only swim time was statistically significant in the dataset, including the outlier (results not shown). When this observation was excluded, none of the variables were statistically significant in univariable analyses. We did not enter water temperature in the multivariable regression model due to the small variability in the measurements.

## 5. Conclusions

We found no association at a group level between body core temperature at the end of the swim leg and swim time, sex, or BMI among participants wearing wetsuits in temperatures of 14.4–16.4 °C. However, one participant, who constituted an outlier in the study, did experience hypothermia and had a low BMI and a long swim time. This finding indicates that participants who share these characteristics may be at risk for hypothermia.

## Figures and Tables

**Figure 1 sports-09-00087-f001:**
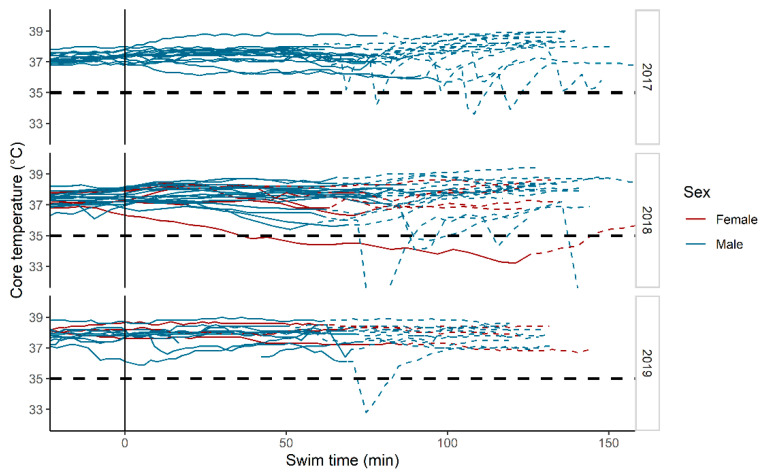
Solid lines show individual temperature recordings during the swim. Dashed lines show temperatures during the first hour after the swim (transition and start cycling). Time 0 is race start (05:00 a.m.).

**Figure 2 sports-09-00087-f002:**
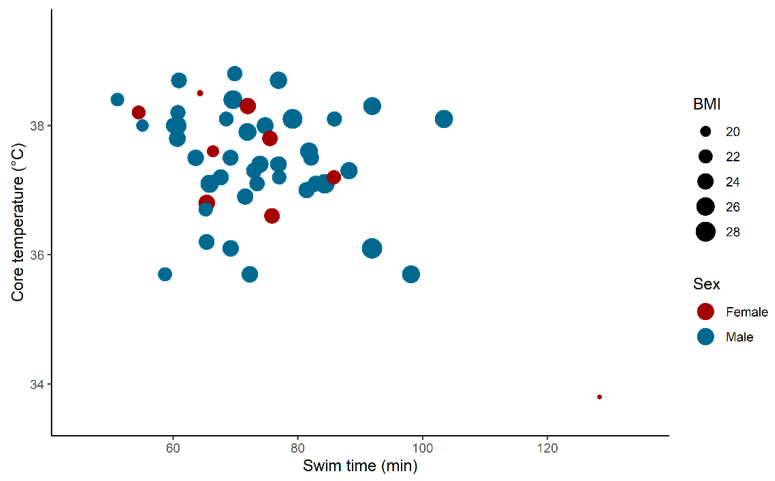
Core temperature at the end of swim vs. swim time. Body mass index (BMI) as point size, sex as color. One female subject with a low BMI and a long swim time (bottom right) constitutes an outlier.

**Figure 3 sports-09-00087-f003:**
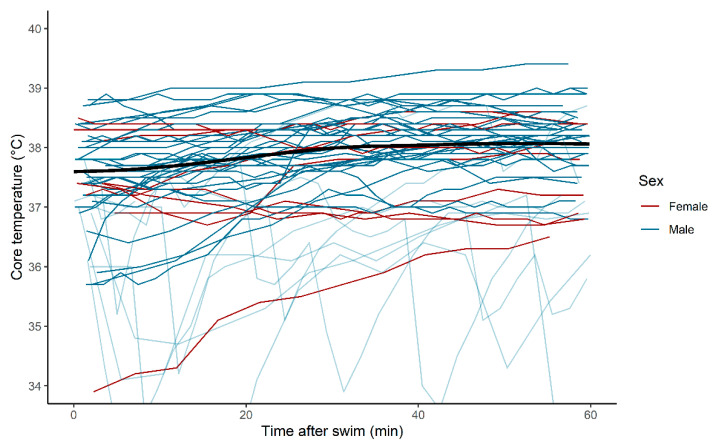
Temperature after the end of the swim. Measurements from subjects with seemingly erroneous readings are faint, and do not contribute to the bold black local polynomial regression (LOESS)—smoothed line.

**Table 1 sports-09-00087-t001:** Demographic, anthropometric, and physiological characteristics and swim time. Data are shown as mean ± (SD) (n = 51).

Characteristic	Mean (SD)
Age (years)	38 (11)
Height (cm)	178 (8)
Weight (kg)	76 (10)
BMI (kg/m^2^)	24 (1.9)
Swim time (min)	72 (13)

**Table 2 sports-09-00087-t002:** Regression coefficients for all subjects. The temperature at the end of the swim is the outcome variable. BMI was not statistically significant and was removed from the model. * denotes interaction effect.

Variable	Estimate	95% CI	*p*-Value
Swim time (min)	−0.060	−0.087 to −0.033	*p* < 0.001
Sex (female reference category)	−3.5	−6.1 to −0.79	*p* = 0.012
Swim time*sex	0.048	0.014 to 0.083	*p* = 0.007

**Table 3 sports-09-00087-t003:** Regression coefficients for subjects except for the outlier. The temperature at the end of the swim is the outcome variable.

Variable	Estimate	95% CI	*p*-Value
Swim time (min)	−0.013	−0.037 to 0.011	*p* = 0.27
Sex (female reference category)	−0.10	−0.79 to 0.58	*p* = 0.76
BMI (kg × m^−2^)	−0.0086	−0.17 to 0.15	*p* = 0.91

## Data Availability

The data presented in this study are available on request from the corresponding author. The data are not publicly available due to data protection regulations.

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
