# Peer review of "Core Temperature during Cold-Water Triathlon Swimming"

_sports, 2021, doi:10.3390/sports9060087_

Round 1

Reviewer 1 Report

First of all, I am not English native speaker, so I hope you understand me well.

The authors do a good job with this study. The discussion section could be refined somewhat. The other sections are better and need minor changes.

Further comments and suggestions:

Line 29: maybe 100,000 participants?

Line 45: maybe 3,800 m? Use the comma correctly in numbers throughout the manuscript.

Line 45-54: I think that this paragraph fits more in a method section. You can summarize it mentioning the fact of shortened swim leg for safety reasons, but you need more consistency in your contextualization of the problem.

Line 55: delete “therefore” or at the beginning of the paragraph.

Line 57: maybe a second purpose? Or the influence of cycling on swim leg?

Line 63: Can you provide a short background for this exclusion criteria? Why the weight of the subject must be below 36.5 kg?

Line 72-73: the sentence “Air temperature for 2017…” seems lack of contextualization here. Why do you measure the temperature of these years?

Line 78: Just for curiosity, how do you know the ingestion of water moment?

Line 80-82: maybe this sentence can help you to improve the introduction section. Try to explain why you use these variables as explanatory.

Line 92: The section is called “Subjects” but you mention “seventy participants”. Change participants to subjects.

Line 94: Now, I understand why you measure the temperature of 2017 and 2018. Specify the years in the study objective.

Line 95: I think that the characteristics of the sample fit more in the Method section in the same way or in a Table.

Line 99: It not the best subheading.  Change it to a more appropriate title.

Line 101: “Temperature measurements during the swim leg and during the first hour after the swim leg…”.

Line 102-104: You mention that only one subject experienced hypothermia in 2018, however, in the Figure 1 there are 1 solid line (red) under the 35º. I think that the Figure 1 can improved. Maybe, more vertical lines for each stage of the event.

Line 136: After swim leg.

Figure 3: What can be due to the peaks of some subjects? Variations of 2 degree per 10min. The outlier was this subject?

Line 154-163: It is necessary to add more information and relation with your results in this paragraph. Maybe the importance of the fat…

Line 172: This affirmation of a slow swimmer…It would be better to delve into the duration of the effort along with the temperature.

Line 175-210: I think that these paragraphs are more a repetition of the results more than a discussion with other manuscripts. Please, improve it.

Author Response

Reviewer 1:

The authors do a good job with this study. The discussion section could be refined somewhat. The other sections are better and need minor changes.

Answer: We agree, and have done some structural changes in the discussion section  

Further comments and suggestions:

Line 29: maybe 100,000 participants?

Answer: We agree and have changed it according to your suggestion. (Page 1, line 29)

Line 45: maybe 3,800 m? Use the comma correctly in numbers throughout the manuscript.

Answer: We agree and have changed the notation according to your suggestion throughout the manuscript.

Line 45-54: I think that this paragraph fits more in a method section. You can summarize it mentioning the fact of shortened swim leg for safety reasons, but you need more consistency in your contextualization of the problem.

Answer: We agree that there is a need for consistency and have added text to clarify. (Page 2, line 1-9)

Line 55: delete “therefore” or at the beginning of the paragraph.

Answer: We agree and have changed it according to your suggestion. (Page 2, line 17)

Line 57: maybe a second purpose? Or the influence of cycling on swim leg?

Answer: We agree that this is a second purpose of this study. The first hour of cycling is included since afterdrop probably will occur while the participants is in the beginning of their bike ride. (Page 2, line 18-20)

Line 63: Can you provide a short background for this exclusion criteria? Why the weight of the subject must be below 36.5 kg?

Answer: The exclusion criteria, namely history of gastrointestinal diseases or surgery, implanted medical devices, scheduled MRI scan within seven days after the race, pregnancy, or weight below 36.5 kg, are the ones listed by the manufacturer of the temperature sensors. This information has now been added to the text. (Page 2, line 29-30)

Line 72-73: the sentence “Air temperature for 2017…” seems lack of contextualization here. Why do you measure the temperature of these years?

Answer: Data were recorded in 2017, 2018 and 2019 and water and air temperature are therefore reported for these years. We agree that this was not clearly written in the “Material and Methods” section and information about this is added to the text. (page 2, line 37-40)

Line 78: Just for curiosity, how do you know the ingestion of water moment?

Answer: In some of the participants, we observed some large and transient drops in temperature that obviously could not be a real drop in core temperature. The pattern is the same as we often observe after drinking cold drinks and we therefore assume that the reason is ingestion of cold water.

Line 80-82: maybe this sentence can help you to improve the introduction section. Try to explain why you use these variables as explanatory.

Answer: Thank you for this suggestion. The introduction section has been rewritten.

Line 92: The section is called “Subjects” but you mention “seventy participants”. Change participants to subjects.

Answer: We agree and have changed it according to your suggestion. (Page 3, line 4)

Line 94: Now, I understand why you measure the temperature of 2017 and 2018. Specify the years in the study objective.

Answer: We agree and have added information about this in the method section. (Page 2, lines 17-18 and 38-39)

Line 95: I think that the characteristics of the sample fit more in the Method section in the same way or in a Table.

Answer: Thank you for this suggestion. We added information about the number of participants to the method section and also presented the demographic data in a table in the beginning of the result section.

Line 99: It not the best subheading.  Change it to a more appropriate title.

Answer: We agree and have changed the subheading to “Core temperature during the swim” and “Core temperature after the swim” for the next paragraph. (Page 4, line 31 and page 6, line 22)

Line 101: “Temperature measurements during the swim leg and during the first hour after the swim leg…”.

Answer: This sentence has been revised (and clarified) as suggested. (Page 6, line 33)

Line 102-104: You mention that only one subject experienced hypothermia in 2018, however, in the Figure 1 there are 1 solid line (red) under the 35º. I think that the Figure 1 can improved. Maybe, more vertical lines for each stage of the event.

Answer: The one solid line in 2018 is this one subject. It is difficult to add a vertical line at the end of the swimming since the swim time differ between the participants. Instead, we have indicated the difference between the time in water with a bold line and after the swim with dotted line. 

Line 136: After swim leg.

Answer: We agree and have changed the subheading to “Core temperature after the swim”. (Page 6, line 22)

Figure 3: What can be due to the peaks of some subjects? Variations of 2 degree per 10min. The outlier was this subject?

Answer: The reason for the drops is probably ingestion of cold drinks. The results are presented so the reader have the opportunity to see how this affect the recordings. The outlier is the subject steadily increasing from approx.. 34ËšC.

Line 154-163: It is necessary to add more information and relation with your results in this paragraph. Maybe the importance of the fat…

Answer: We agree and have added text to clarify. (Page 7, Line 13-24)

Line 172: This affirmation of a slow swimmer…It would be better to delve into the duration of the effort along with the temperature.

Answer: The reason for the large drop in core temperature in this subject is probably a combination of low BMI and slow swimming. We have added the swim time for this subject to the text to make it easier for the readers to make their own reflections. (Page 7, line 29)

Line 175-210: I think that these paragraphs are more a repetition of the results more than a discussion with other manuscripts. Please, improve it.

Answer: We agree, and have revised the text.

Reviewer 2 Report

Thank you for the opportunity to read and review this manuscript. It was very well-written and a pleasure to read. Below are some comments to consider for the manuscript.

I think that it would be worthwhile to discuss the generalizability of the findings to other long-distance triathlons. The Norseman race is unique, for many reasons, and thus not representative of a “standard” long distance triathlon. I’ve seen other research publications use the term “extreme” to differentiate this type of race (and others like it eg. Swissman) from the other iron-distance races.

Differences with standard long-distance triathlon are worth discussing. For example, the swimmers can wear neoprene socks and caps, which is not standard and worth mentioning. These may have reduced the body temperature drop.  

Also, it should be mentioned that Norseman would possibly attract a more experienced level of triathlete. This could be demonstrated by comparing medianNorseman swim time to the median swim time for other long distance races. 

Since metabolic rate and hence heat production is tied to exercise intensity, I was surprised that exercise intensity was not mentioned in the manuscript.  When comparing studies, it makes sense to give some indication of the relative exercise intensity in the studies.

As for the discussion of afterdrop, my understanding is that it occurs on transition to a warmer environment.  In Norseman, this may not be the case, as the participants transition from swimming in a wetsuit to cycling in the brisk early morning with usually just a triathlon suit on.  Since afterdrop relates to return of peripheral cool blood to the core once cutaneous vascular beds reopen, this might not be expected in the initial phases of the Norseman bike ride, and certainly would not be as abrupt as in typical studies of hypothermia and rewarming.  

The findings are rather limited, likely due in part to the relatively warm temperatures in the fjord during the years of study.  However, the authors do a good job of discussing the findings that they do have.  

Author Response

Reviewer 2:

Thank you for the opportunity to read and review this manuscript. It was very well-written and a pleasure to read. Below are some comments to consider for the manuscript.

I think that it would be worthwhile to discuss the generalizability of the findings to other long-distance triathlons. The Norseman race is unique, for many reasons, and thus not representative of a “standard” long distance triathlon. I’ve seen other research publications use the term “extreme” to differentiate this type of race (and others like it eg. Swissman) from the other iron-distance races.

Answer: Thank you for this good comment. We have included more information about the different races in the introduction.

Differences with standard long-distance triathlon are worth discussing. For example, the swimmers can wear neoprene socks and caps, which is not standard and worth mentioning. These may have reduced the body temperature drop.  

Answer: We agree that this is a difference that is worth mentioning in the discussion. We have edited the text accordingly. (page 8, line 22)

Also, it should be mentioned that Norseman would possibly attract a more experienced level of triathlete. This could be demonstrated by comparing median Norseman swim time to the median swim time for other long distance races. 

Answer: Thank you for bringing up this point. This is a very interesting comment, and something that we will probably look more into. We are however uncertain if the participants are in fact more experienced, as the large majority of participants in the race are drawn from a lottery. We will respectfully leave this out for the present manuscript, hope to look into this in future studies.

Since metabolic rate and hence heat production is tied to exercise intensity, I was surprised that exercise intensity was not mentioned in the manuscript.  When comparing studies, it makes sense to give some indication of the relative exercise intensity in the studies.

Answer: Thank you for this comment. We have not been able to collect heart rate data or other measures of exercise intensity during these studies. This is however something that we want to follow up in further studies.

As for the discussion of afterdrop, my understanding is that it occurs on transition to a warmer environment.  In Norseman, this may not be the case, as the participants transition from swimming in a wetsuit to cycling in the brisk early morning with usually just a triathlon suit on.  Since afterdrop relates to return of peripheral cool blood to the core once cutaneous vascular beds reopen, this might not be expected in the initial phases of the Norseman bike ride, and certainly would not be as abrupt as in typical studies of hypothermia and rewarming.  

Answer: We agree, and have added text to clarify according to your comment. (Page 8, Line 13-25)

The findings are rather limited, likely due in part to the relatively warm temperatures in the fjord during the years of study.  However, the authors do a good job of discussing the findings that they do have.  

Answer: We thank the reviewers for their valuable comments, and hope the manuscript has been improved by the revision.